# The Impact of Different Cultivation Systems on the Content of Selected Secondary Metabolites and Antioxidant Activity of *Carlina acaulis* Plant Material

**DOI:** 10.3390/molecules25010146

**Published:** 2019-12-30

**Authors:** Maciej Strzemski, Sławomir Dresler, Ireneusz Sowa, Anna Czubacka, Monika Agacka-Mołdoch, Bartosz J. Płachno, Sebastian Granica, Marcin Feldo, Magdalena Wójciak-Kosior

**Affiliations:** 1Department of Analytical Chemistry, Medical University of Lublin, Chodźki 4a, 20-093 Lublin, Poland; i.sowa@umlub.pl; 2Department of Plant Physiology and Biophysics, Institute of Biological Science, Maria Curie-Skłodowska University, Akademicka 19, 20-033 Lublin, Poland; slawomir.dresler@poczta.umcs.lublin.pl; 3Department of Plant Breeding and Biotechnology, Institute of Soil Science and Plant Cultivation, State Research Institute, Czartoryskich 8 St., 24-100 Puławy, Poland; annacz@iung.pulawy.pl (A.C.); magacka@iung.pulawy.pl (M.A.-M.); 4Department of Plant Cytology and Embryology, Institute of Botany, Faculty of Biology, Jagiellonian University, Gronostajowa 9 St. 30-387 Cracow, Poland; bartosz.plachno@uj.edu.pl; 5Department of Pharmacognosy and Molecular Basis of Phytotherapy, Faculty of Pharmacy, Medical University of Warsaw, Banacha 1 St., 02-097 Warsaw, Poland; sgranica@gmail.com; 6Department of Vascular Surgery, Medical University of Lublin, Staszica 11 St., 20-081 Lublin, Poland; marcin.feldo@umlub.pl

**Keywords:** *Carlina acaulis*, callus tissue, depsides, pentacyclic triterpenes, antioxidant activity, hydroponic, soil cultivation

## Abstract

Roots and leaves of *Carlina acaulis* L. are still used in ethnomedicine in many European countries; however, the limited occurrence of the plants and protection of this species necessitate a search for alternative ways for obtaining this plant material. In this study, in vitro cultures, hydroponic cultures, and field cultivation were applied to obtain the *C. acaulis* plant material. Its quality was evaluated using antioxidant activity tests and high performance liquid chromatography analysis. Our study showed that the antioxidant activity and the content of chlorogenic and 3,5-di-caffeoylquinic acid in roots of plants cultivated in hydroponics and field conditions were comparable. However, the amount of carlina oxide was significantly higher in plants from the field. The flavonoid content in leaves obtained from both cultivation systems was at the same level; however, the antioxidant activity and the content of the investigated metabolites were higher in the soil cultivation system. The callus line exhibited high differentiation in phytochemical compositions depending on the treatments and medium compositions.

## 1. Introduction

*Carlina acaulis* L. (Asteraceae) has great importance in traditional medicine. The root of this plant was used in ancient times through the Middle Ages until the 19th century. Monographs of this root have been found in many pharmacopoeias. A detailed study on the historical uses of this raw material was published by Strzemski et al. [1].

Nowadays, the medical application of *C. acaulis* is limited to folk medicine. Both roots and aboveground parts of the plant are used as diaphoretic, diuretic, anthelmintic, and antibacterial drugs in ethnomedicine in Italy [2,3], Montenegro, Bosnia and Herzegovina, Serbia [4], and Macedonia [5]. The biological activity of extracts from Carlina plants is associated with the presence of such compounds as pentacyclic triterpenes: oleanolic (OA) and ursolic acids (UA) [6], chlorogenic acid (ChA), and 3,5-di-caffeoylquinic acid (3,5CQA) [7,8]; flavonoids: vitexin, orientin, homoorientin and schaftoside [9,10]; and carlina oxide (CO) [10,11,12,13,14].

Although phytochemical and pharmacological studies of extracts from leaves and roots of *C. acaulis* confirm the therapeutic potential of this plant, the raw material was withdrawn from phytotherapy, probably due to the limited presence of *C. acaulis* in natural habitats. Moreover, *C. acaulis* is currently a protected species in many countries [15]. Therefore, it is necessary to search for alternative methods for obtaining this plant material. In vitro culture is widely believed to be one of such methods. Compared to conventional cultivation, it ensures stable conditions irrespective of the vegetation season and environmental factors [16]. Callus tissues obtained in in vitro culture, after a period of stabilization, may be a basis for the production of metabolites in cell suspension culture. Moreover, through proper selection of nutrient components, it is possible to influence plant biomass production and metabolic pathways, which may lead to induction of biosynthesis of specific compounds. However, the contents of some metabolites are often much lower in callus tissues than in those obtained from plants growing in natural conditions [17]. Thus, it is not always possible to obtain equivalent material in in vitro cultures. Moreover, in vitro cultivation requires the experience and high expenditure of labor associated with passaging of the cultures. Another way of producing plant material is hydroponic culture. In hydroponic cultures, it is possible to use a suitable nutrient composition and eliminate deficiencies of some elements that may occur in soil. In addition, hydroponic crops eliminate the problem of poor soil structure and reduce the risk of pathogens and plant pests present in soil. As a result, much faster plant growth is achieved. Hydroponic cultivation can also be carried out in controlled conditions of temperature, humidity, and day length [18]. In addition, in some cases, hydroponic cultivation can contribute to an increase in the content of biologically active compounds in plants, compared to soil cultivation [19]. The main disadvantage of hydroponics is difficulty with performing the long lasting cultivation and several cycles are needed to obtain the appropriate biomass. Soil cultivation is a classic way of obtaining plant material. Its advantage is the possibility of production without the need for a specialized laboratory and the obtaining of high biomass at relatively low cost. However, this type of production may have an impact on the natural environment, due to, e.g., introduction of species that are not native to a given environment. Moreover, the quality and abundance of crop strongly depends on environmental factors and they are hardly to predict.

The objective of our research was to obtain *C. acaulis* material using different cultivation systems such as conventional field cultivation, hydroponics, and in vitro cultures. The content of selected biologically active compounds, i.e., phenolic acids, triterpenic acids, and CO, was assessed in the plant material. Moreover, the antioxidant activity shown by TPC, DPPH, and ABTS assays and flavonoid content were evaluated.

## 2. Results and Discussion

*C. acaulis* is a valuable plant with confirmed biological activity. Our previous investigation revealed the presence of high amounts of ChA, OA, UA, and CO [6,8,12,20]. However, due to the limited occurrence of *C. acaulis* and protection of this species, collection of plant material from natural sites is difficult; thus, the search for alternative approaches is important for the widespread use of this plant.

### 2.1. Soil and Hydroponic Cultivation

The soil cultivation of *C. acaulis* lasted two years in order to obtain plant material for comparison with that investigated in our previous research. The average fresh weight of the plant was 153.5 ± 45.9 and 32.7 ± 6.0 g of leaves and roots, respectively.

The hydroponic cultivation lasted 60 days, and the weight of the plant material was 15.7 ± 4.8 and 4.2 ± 1.3 g of leaves and roots, respectively. Cultivation of *C. acaulis* plants in hydroponics for a longer time was difficult due to technical problems such as intensive growth of algae in the medium.

Note that the plants obtained after one year of field cultivation were smaller in size than those after 60 days growing in hydroponics. However, after two years the biomass of plant cultivated in soil was significantly higher comparing to 60 days old hydroponic plants. Therefore, it can be concluded that the cultivation in hydroponics resulted in faster growth of *C. acaulis* plants than the soil cultivation and it was possible to carry out several cycles of cultivation at relatively short time to obtain the appropriate amount of biomass. Photographs of examples of plants obtained in hydroponic and soil cultivation are shown in Appendix A.

### 2.2. In Vitro Plant Material

The seeds germinated after seven days and formed seedlings with different morphology. Some plants had tomentose leaves while others had glabrous forms. Callus tissues were induced from the leaves and roots of both types of plants. As can be seen in Table 1, the type of the explant and the composition of the medium influenced the efficiency of callus production.

Leaves showed to be better explants than roots, as the callus tissues induced from leaf explants were generally more abundant. The combinations of phytohormones were important for the effectiveness of callus induction as well. The combination of IBA (indole-3-butyric acid) with KIN (kinetin) showed to be more effective than the combination with 2iP (6-(γ,γ-dimethylallylamino)purine) and resulted in formation of a more abundant callus from leaf fragments on medium B than those on medium A. The combination of 2,4-D (2,4-dichlorophenoxyacetic acid) with KIN or BA (6-benzylaminopurine) in the equal proportion favored the callus growth as well. On the contrary, the addition of TDZ (thidiazuron) resulted in poor production of the callus tissue regardless of the plant organ used as an explant. Organogenesis occurred in some cases, i.e., the leaf and root fragments on medium A as well as the root explants on medium B regenerated into both roots and shoots (Appendix A). Moreover, single roots regenerated from the callus obtained on medium C from leaf fragments were also observed.

The plant morphology had a slight impact on the effectiveness of callus induction. It was only noticed that the root explants of the tomentose plants formed more abundant tissue on the medium enriched with IBA and KIN than those from the glabrous individuals. In turn, on medium with 2,4-D and KIN, explants originating from the glabrous leaves produced more callus tissue than those from the tomentose ones. The plant morphology did not affect the physical characteristics of the calluses obtained and their structure depended on the type of medium. Hard and compact callus tissue was formed from fragments of plants placed on medium A, whereas hard tissue with grainy structure was observed on medium C (Appendix A). A similar tissue was formed on the leaf explants on medium D and a tissue with looser structure was formed on the root fragments. A friable and grainy callus was observed on medium B, whereas the callus formed on medium E had the loosest structure. As can be seen in Appendix A, the callus tissues had cream, beige, or light green color.

Although the initial amount of callus was moderate or even abundant, in some cases, necrotic changes appeared after transferring it onto the fresh medium. This limited callus growth, and collection of a sufficient amount of material for phytochemical analyses was impossible. This was observed for callus tissues obtained from roots on medium A, roots of plants with tomentose leaves on medium C, and those induced from glabrous leaves on medium B.

### 2.3. Scanning Electron Microscopy

An extracellular matrix (ECM) was observed on the surface of cells of all examined calluses (Figure 1A–F). The ECM formed a membranous layer in TLD (callus from leaves of plants with tomentose leaves, growing on medium D) callus (Figure 1A,B) or complex reticulate network in GRC (callus from roots of plants with glabrous leaves growing on medium C) and GRD (callus from roots of plants with glabrous leaves growing on medium D) calluses (Figure 1C,D). The external ECM appeared as a membranous layer, which covered the fibrillar structure in the GRC callus. The TLD line also exhibited membranous ECM transformed into fibrillar ECM.

The extracellular matrix (ECM) was observed in callus cultures of various species to play a role in cell-to-cell interaction [16,21].

### 2.4. Spectroscopic Analysis

Plants are a rich source of polyphenols, which act as antioxidants, and are therefore helpful in preventing oxidative stress [22]. For this reason, the antioxidant activity and flavonoid content in the extracts from the *C. acaulis* material were evaluated (Table 2). The differences in FC (flavonoid content), TPC (antioxidant activity evaluated in tests with Folin–Ciocalteu reagent), and DPPH between roots from the hydroponics and the field cultivation were statistically insignificant. The values obtained were 2.92–3.69, 18.73–19.90, and 4.72–5.24 mg∙g^−1^ of dry plant material, respectively. A higher value was observed only in the case of ABTS in plants cultivated in soil. In the case of leaves, significantly higher values of TPC, DPPH, and ABTS were determined for field plants, compared to the hydroponic cultivation, whereas the FC values were at the same level. The FC content in the leaves of the soil and hydroponic plants were 34.67 mg∙g^−1^ dry weight (DW) and 38.26 mg∙g^−1^ DW, respectively.

High differentiation was obtained in the callus line with values in the range of 0.05–2.18 for FC, 5.9–35.89 for TPC, 2.18–7.35 for DPPH, and 8.56–15.2 for ABTS, depending on the type of medium, and no tendency was observed taking into account the type of the explants.

Generally, significantly lower values were obtained for the callus lines than for plants from the soil cultivation; surprisingly, high TPC values were determined for the TRE (callus from roots of plants with tomentose leaves growing on medium E) and GRE (callus from roots of plants with glabrous leaves growing on medium E) lines.

### 2.5. HPLC-PDA Analysis

The results of the chromatographic determination of metabolites in the investigated plant materials are presented in Table 3. The chromatographic analysis revealed the presence of two major metabolites (ChA and 3,5CQA) in the extracts of the calluses tested. The calluses of all lines except GRD contained these compounds. The ChA content ranged from 0.13 (TLE line: callus from leaves of plants with tomentose leaves growing on medium E) to 8.47 (GRC line) mg∙g^−1^ dry plant material (DW). The 3,5CQA content ranged from 0.08 to 2.97 mg∙g^−1^ DW in the same lines. High levels of phenolic acids were found in the GLD (callus obtained from leaves of plants with glabrous leaves growing on medium D) and TLD lines. The callus of the GLD line contained 6.50 mg∙g^−1^ DW ChA and 2.93 mg∙g^−1^ DW 3,5CQA, whereas the TLD callus line had 6.35 mg∙g^−1^ DW ChA and 2.02 mg∙g^−1^ DW 3,5CQA. Similar to the antioxidant activity assay, no tendency was observed taking into account the type of the explants.

It should be noted that the leaves of *C. acaulis* plants obtained from the field cultivation contained more than 10 mg∙g^−1^ DW ChA. Such a high content of this compound was not found in any callus lines or in plants from the hydroponic cultures (Table 3); however, the value for some lines was quite high (ca 8.4 mg∙g^−1^), which indicates the potential for further investigations. Substantially larger amounts of ChA were found in the leaves than roots, while an inverse relationship was noted for 3,5CQA. The content of 3,5CQA determined in the roots of plants growing in the field and hydroponic cultures was lower than in the GRC callus line. Therefore, it can be concluded that the high accumulation of 3,5CQA is typical for root tissues and that callus cultures can be a valuable material for isolation of this compound.

One of the richest sources of chlorogenic acids are coffee seeds. They contain up to 59 mg∙g^−1^ of these compounds, including ChA and 3,5CQA [23]. Although the *C. acaulis* callus lines and plants are a less efficient source of chlorogenic acids than coffee seeds, they should be regarded as a relatively ChA- and 3,5CQA-rich material. This is very important in terms of the health-promoting properties of these compounds, which have been confirmed in numerous studies [24,25,26].

The chromatographic analysis of pentacyclic acids revealed the presence of a low amount of OA and UA in the extracts from the majority of calluses (Table 3) except those from the TLD, GRB (callus obtained from roots of plants with glabrous leaves growing on medium B), and GRD lines. The content of OA ranged from 0.03 (TRD line: callus obtained from roots of plants with tomentose leaves growing on medium D) to 0.23 (GLE lines: callus obtained from leaves of plants with glabrous leaves growing on medium E) mg∙g^−1^ DW, and UA was in the range from 0.03 (GRC line) to 0.29 (GLE line) mg∙g^−1^ DW. Although only low amounts of triterpenic acids were found in the callus tissues obtained from the roots, their occurrence is interesting given the absence of these compounds in the roots of plants from the natural state [6]. The reduction of the UA and OA concentration in comparison to naturally growing plants is characteristic for the in vitro model [27,28,29].

Generally, the content of OA and UA in plants cultivated in both the field and hydroponics was higher than that determined in *Carlina* plants growing in the natural environment. The content of UA and OA in the leaves of such plants was approx. 2.8 mg∙g^−1^ DW and 0.5 mg∙g^−1^ DW, respectively [6]. Our current study showed that the content of these compounds in the leaves of the field crops reached 2.05 mg∙g−1 DW and 5.57 mg∙g^−1^ DW of OA and UA, respectively. The leaves of plants growing in hydroponics contained 1.20 mg∙g^−1^ DW of OA and 3.64 mg∙g^−1^ DW of UA (Table 3).

The greatest differentiation between the tested materials was found in the content of CO (Table 3 and Figure 2). This polyacetylene is regarded as the main biologically active component of *C. acaulis* roots. The roots of plants growing in soil contained 8.73 mg∙g^−1^ DW of this compound, while the roots of plants grown in hydroponics exhibited only 0.14 mg∙g^−1^ DW. No CO was found in any of the tested calluses. This can be explained by the fact that the essential oil is collected in specialized glands and not in cells plants [30]. The large difference in the CO content between the soil-grown plants and hydroponic plants may be related to both different growth conditions and the age of the plant [31,32]. Taking into consideration the latest research on insecticidal activity of CO, which showed that CO is a very effective insecticide [14], it should be assumed that this compound can be used for plant protection in the future. CO is found in only a few plant species and soil cultivation of *C. acaulis* can be a promising way to obtain it. Hydroponic cultivation may potentially be considered as a source of CO however, further research is required.

For clarity of the results, heat maps were created for all standardized data (Figure 3). The parameters determined for each individual are represented by colors (dark green: very low value; dark red: very high value). Based on the phytochemical data, hierarchical cluster analysis (HCA) was performed. The HCA of the shoot data revealed grouping of the raw materials in two main clusters. One cluster consisted of the roots and leaves of plants from hydroponic and soil cultivation and the TRE, TLD, GRC, and GLD callus lines. The other cluster comprised all the other callus lines. The HCA analysis based on the quantitative data of individual analytes also revealed the existence of two main clusters. One cluster consisted of triterpenic acids (OA and UA), and the other cluster included depsides (ChA, 3,5CQA), CO, FC, and antioxidant activity (DPPH, ABTS, and TPC).

### 2.6. Principal Component Analysis

The PCA analysis of the variables showed that the first three components explained 33.1, 23.0 and 17.5% of the total variability, respectively (Figure 4). The first compound is responsible for the separation of the individuals into two major groups, which corresponds with the HCA results. It was noted that the first PC was positively correlated (correlation coefficient above 0.5) with DPPH, ChA, phenolic content, flavonoid content, and 3,5CQA, whereas the second PC was negatively correlated with oleanolic and ursolic acids and ABTS. The second PC facilitated separation of samples with the higher content of triterpene acids such as GLE, GLD, TRB (callus obtained from roots of plants with tomentose leaves growing on medium B), and GRE from those without or low content thereof, i.e., SH (shoot of a plant grown in hydroponic culture), RS (root of a plant grown in soil culture), GRD, RH (root of a plant grown in hydroponic culture), GRB, SS (shoot of a plant grown in soil culture), TLE, or TLD (Figure 4A). In turn, the third PC is strongly negatively correlated with CO, 3,5CQA, and TPC. It facilitated separation of samples with high content of CO (RS) or with higher accumulation of TPC and 3,5CQA (GRC, TRE) (Figure 4B).

## 3. Materials and Methods

### 3.1. Reference Standard and Chemicals

Murashige & Skoog Medium Vitamin Mixture (MS), auxins: IBA (>98.0%), 2,4-D (>96%), and cytokinins: KIN (>98%), BA (>99%), 2iP (>98%), and TDZ (>95%) were purchased from Duchefa Biochemie B.V. (Haarlem, the Netherlands). Acetonitrile (ACN) and trifluoroacetic acid (TFA) of at least pro-analysis-grade were supplied by Merck (Darmstadt, Germany). 2-azino-bis(3-ethylbenzthiazoline-6-sulphonic acid) (ABTS), 2,2-diphenyl-1-picrylhydrazyl (DPPH), Folin–Ciocalteu reagent, trolox, aluminum chloride, and standards of analytes: ChA (≥95.0%), 3,5CQA (≥95.0%), rutin (≥94.0%), UA (≥90.0%), and OA (≥97.0%) were purchased in Sigma (St. Louis, MO, USA). CO standard was obtained by distillation of *C. acaulis* roots in the Deryng apparatus. The purity of the standard (96.2%) was evaluated using GC-MS, and the identity was confirmed by IR, Raman, and NMR spectroscopy according to the methodology published previously [13]. Water was deionized and purified using Ultrapure Millipore Direct-Q^®^ 3UV-R (Merck, Darmstadt, Germany).

### 3.2. Plant Material

The seeds of *C. acaulis* (voucher specimen no. 2005A) from plants growing in soil were obtained from the Botanical Garden of Maria Curie-Skłodowska University in Lublin, Poland (51°16′ N, 22°30′ E). They were used in the cultivation experiments and in vitro cultures.

#### 3.2.1. Field Cultivation

The seeds were placed in soil and seedlings were grown in a greenhouse for 60 days. The field plantation was established at the beginning of April 2017. Plants were grown at 40 × 40 cm spacing. The field was not fertilized and no plant protection products were used. The plantation was weeded mechanically. The plants were collected in the second half of July 2018 in the vegetative phase of growth.

#### 3.2.2. Hydroponic Cultivation

For the hydroponic cultivation, the seeds were germinated on the surface of garden soil and 10-day old seedlings were transferred into polyethylene pots with the soil. After 28 days, the plants were carefully washed with distilled water and transferred into pots (one plant per pot) with 0.5 L of half strength Hoagland’s medium [29]. The solutions were continuously aerated, evapotranspiration loss was replenished daily with water, and the medium was renewed every 14 days. Cultivation was carried out in a growth chamber at 18/25 °C (8/16 h dark/light photoperiod) under light-emitting diodes at photosynthetic photon flux density of 150 µmol m^−2^ s^−1^ and relative humidity of 60–65%. The plants were harvested after 60 days of cultivation in hydroponics.

#### 3.2.3. In Vitro Cultures

Seeds were surface-sterilized first with 70% (*v*/*v*) ethanol for 30 s, soaked in a 20% (*v*/*v*) commercial solution of NaClO for 20 min, and rinsed four times with sterile distilled water [33]. Next, the seeds were sown and germinated in Petri dishes on MS medium [34] solidified with 0.7% (*w*/*v*) plant agar. The seedlings were transplanted into transparent plastic vessels on MS medium with 0.8% (*w*/*v*) agar and maintained under 16/8 h (light/dark) photoperiod at 24 °C. After 8 weeks of growth, plantlets were used as a source of explants for the induction of callus tissue. Fragments of leaves with an approx. 0.5-cm^2^ area and ~1-cm-long roots were placed in Petri dishes on MS medium enriched with vitamins and phytohormones, i.e., cytokinins: BA, KIN, 2iP, and TDZ, and auxins: IBA and 2,4-D. Five variants of the MS media were applied: A—0.5 mg∙L^−1^ IBA and 2 mg∙L^−1^ 2iP; B—0.5 mg∙L^−1^ IBA and 2 mg∙L^−1^ KIN; C—1 mg∙L^−1^ 2,4-D and 1 mg∙L^−1^ KIN; D—1 mg∙L^−1^ 2,4-D and 1 mg∙L^−1^ BA; E—2 mg∙L^−1^ 2,4-D and 1 mg∙L^−1^ TDZ. The compositions of the medium were selected based on a preliminary study (data not shown).

The Petri dishes with plant explants were maintained in the dark at 24 °C. After 8 weeks, callus tissues were visually assessed in terms of abundance and color and their structure was observed. Callus tissues were cut off from the explants, transferred onto fresh medium, and subcultured several times every 4–6 weeks until homogeneous tissue was obtained and multiplied.

### 3.3. Scanning Electron Microscopy

The material for SEM was fixed in a mixture of 2.5% glutaraldehyde with 2.5% formaldehyde in 0.05 M cacodylate buffer pH 7.2 (Sigma-Aldrich, Saint Louis, MO, USA) for four days, washed with buffer, dehydrated, and subjected to critical-point drying using CO_2_. The material was then sputter-coated with gold and examined at an accelerating voltage of 20 kV using a Hitachi S-4700 scanning electron microscope (Hitachi, Tokyo, Japan), which is available in the Institute of Geological Sciences, Jagiellonian University in Kraków, Poland.

### 3.4. Standards and Sample Preparation

Standards of analytes were dissolved in methanol (the final concentration of the stock solutions was ~100 µg∙mL^−1^) and diluted to appropriate concentrations. The plant material was frozen at a temperature of −80 °C, dried by lyophilization, and pulverized. The powder (0.1000 g) was extracted three times with a fresh portion of methanol (2, 2, and 1 mL) in an ultrasonic bath for 15 min at ambient temperature. The extracts were combined, centrifuged at 10,000 *g* for 10 min, and filled up to 5 mL in volumetric flasks. Directly before the HPLC analysis, the samples were filtered through a 0.22 µm nylon membrane filter.

### 3.5. Spectroscopic Analysis

The antioxidant activity and flavonoid content (FC) were determined according to the methodology published previously [35]. The assay was carried out using a Bio-Rad Benchmark Plus microplate spectrometer (Bio-Rad, Hercules, CA, USA). The antioxidant capacity of the extracts was determined with the use of ABTS and DPPH and expressed as a trolox equivalent per gram of dry weight (mg∙g^−1^ DW). The total phenolic content (TPC) was established using the Folin–Ciocalteu reagent, and the flavonoid content (FC) was analyzed based on the reaction with aluminum chloride. TPC and FC were expressed as an equivalent of gallic acid (mg GAE∙g^−1^ DW) and rutin (mg RUE∙g^−1^ DW), respectively.

### 3.6. HPLC-PDA Analysis

The analyses were performed on a VWR Hitachi Chromaster 600 chromatograph with a PDA detector and EZChrom Elite software (Merck, Darmstadt, Germany). The RP18 reversed-phase column (RP18e LiChrospher 100, Merck, Darmstadt, Germany) (25 cm × 4.0 mm i.d., 5 µm particle size) was used for all analyses. To determine the chlorogenic acids, a mixture of ACN with 0.025% of TFA (solvent A) and water with 0.025% of TFA (solvent B) was used. The compounds were separated by gradient elution with the following program: 0–25 min A 8%, B 92%; 25–60 min A 8–37%, B 92–63%. The flow rate was 1 mL/min and the temperature of the thermostat was 25 °C. Triterpenic acids were determined in an isocratic system according to the methodology published previously [6]. A mixture of ACN, water, and phosphoric acid aqueous solution at a concentration of 1% (75:25:0.5, *v*/*v*/*v*) was used as a mobile phase. The flow rate was 1.0 mL/min, and the temperature of the thermostat was 10 °C. CO was determined according to methodology published previously [12]. Isocratic elution was carried out using a mixture of ACN and water (65:35, *v*/*v*) at a flow rate of 1.0 mL/min and a temperature of 20 °C. Data was collected between from 190 to 400 nm. The identity of compounds was established by comparison of retention times and spectra with corresponding standards. The quantitative analysis was performed at λ = 324 nm, 200 nm, and 249 nm for phenolic acids, triterpenic acids, and CO, respectively.

### 3.7. Statistical Analysis

Data were analyzed using STATISTICA ver.10 (StatSoft, Inc., Tulsa, OK, USA). Differences between the means across treatment groups were evaluated using a one-way analysis of variance (ANOVA) followed by Tukey’s test. The differences between the treatments were determined with Tukey’s honest significant difference test at the 0.05 probability level. The principal component analysis (PCA) was performed based on mean values for each treatment (*n* = 3). The relationship between the treatments was visualized as a dendrogram constructed on the basis of Euclidean distance and Ward’s method. The heat map was created based on standardized mean values for each treatment (*n* = 3).

## 4. Conclusions

In the present study, the *C. acaulis* plant material was obtained using three methods: in vitro cultures, hydroponics, and field cultivation and compared in terms of antioxidant activity and the content of selected secondary metabolites such as phenolic acids, OA and UA, and CO. The chromatographic analysis revealed the presence of two major phenolic acids: ChA and 3,5QA in most samples. CO was found only in extracts from the roots of plants cultivated in soil and hydroponics; however, its amount differed significantly and was 8.73 and 0.14 mg∙g^−1^ of dry plant material, respectively. Triterpenic acids were found in the leaves of plants from the field and hydroponic cultivation. Substantially lower amounts of these compounds were detected in most of callus tissues. Moreover, the antioxidant activity and the ChA and 3,5QA content in the roots of plants cultivated in hydroponics and field conditions were similar. In turn, the antioxidant activity of leaf extracts and the content of the investigated metabolites were higher in plants from the soil cultivation. High differentiation of results was observed in the case of callus tissue depending on the type of medium. In some samples, the amount of ChA was comparable and that of 3,5QA was even higher than in the case of the field cultivated plants. It can therefore be assumed that *C. acaulis* callus cultures can be a valuable source of 3,5QA; however, soil cultivation allows obtaining high biomass and raw material relatively rich in CO as well as triterpenic and phenolic acids. Despite the possibility of conducting many cultivation cycles per year, hydroponic cultures cannot provide raw material rich in CO.

## Figures and Tables

**Figure 1 molecules-25-00146-f001:**
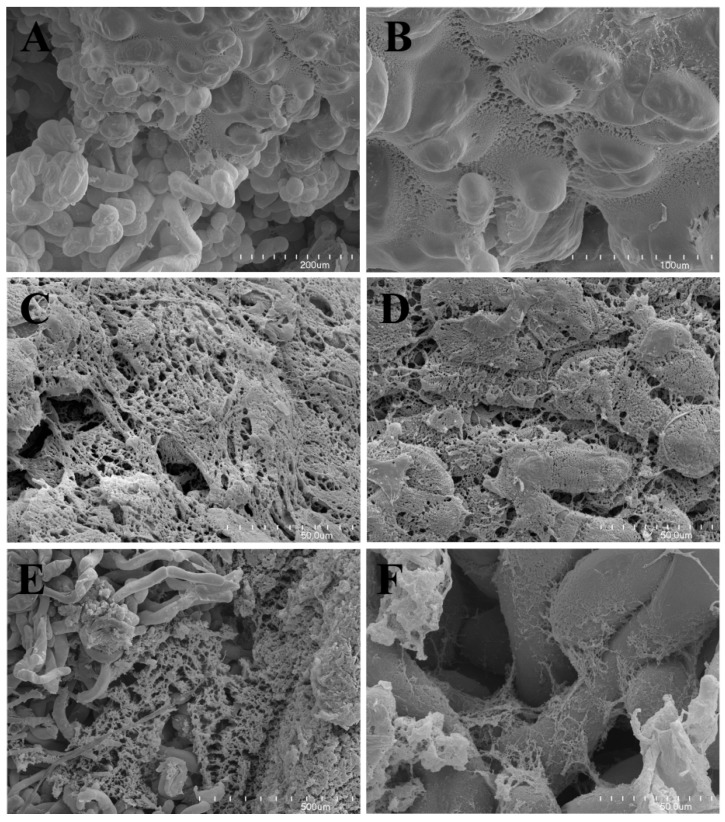
(**A**) SEM observation of the TLD callus; note the extracellular matrix covering a part of the callus cells. (**B**) SEM observation of the TLD callus; transition of the membranous ECM to the fibrillar form. (**C**) SEM observation of the GRC callus; note the ECM forming a complex reticulate network. (**D**) SEM observation of the GRC callus; note the fragmented external membranous ECM persisting only in fragments, thus the internal fibrillar ECM was well exposed. (**E**,**F**) SEM observation of the GRD callus with the external ECM removed. Note the delicate network of ECM between the callus cells. SEM: scanning electron microscopy. ECM: extracellular matrix, TLD: callus from leaves of plants with tomentose leaves growing on medium D. GRC: callus from roots of plants with glabrous leaves growing on medium C. GRD: callus from roots of plants with glabrous leaves growing on medium D.

**Figure 2 molecules-25-00146-f002:**
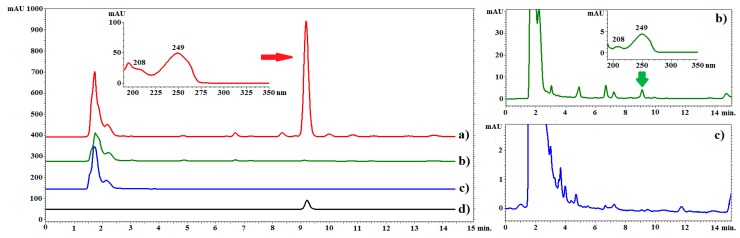
Example chromatograms of *C. acaulis* extracts and the reference compound. Line (**a**): chromatogram of the root extract of plants growing in soil; line (**b**): chromatogram of the extract from the roots of plants grown in hydroponics, line (**c**): chromatogram of the extract from callus, line (**d**): chromatogram of the standard solution (carlina oxide). LiChrospher 100 column, acetonitrile–water (65:35 *v*/*v*), flow rate of 1.0 mL/min at 20 °C. λ = 249 nm.

**Figure 3 molecules-25-00146-f003:**
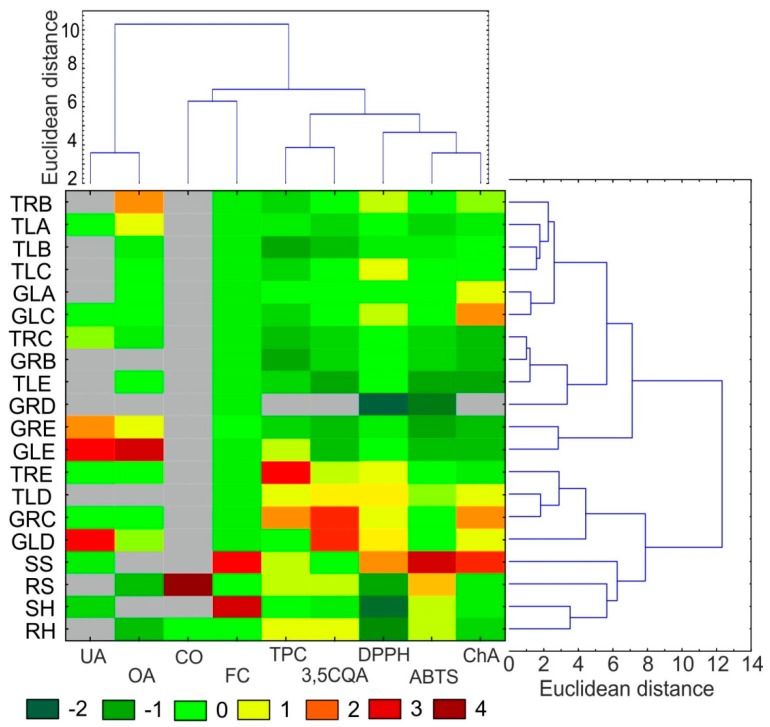
Heat map visualization of changes in the content of selected biological compounds and antioxidant activities shown in the columns for different materials from *C. acaulis*. The colors range from green (low abundance) to deep red (high abundance); * < LOD: under limit of detection (gray color). The right and top side of the figure shows hierarchical clustering dendrograms based on the Euclidean distances with the Ward’s method. TRB: callus from roots of plants with tomentose leaves growing on medium B, TLA: callus from leaves of plants with tomentose leaves growing on medium A, TLB: callus from leaves of plants with tomentose leaves growing on medium B, TLC: callus from leaves of plants with tomentose leaves growing on medium C, GLA: callus from leaves of plants with glabrous leaves growing on medium A, GLC: callus from leaves of plants with glabrous leaves growing on medium C, TRC: callus from roots of plants with tomentose leaves growing on medium C, GRB: callus from roots of plants with glabrous leaves growing on medium B, TLE: callus from leaves of plants with tomentose leaves growing on medium E, GRD: callus from roots of plants with glabrous leaves growing on medium D, GRE: callus from roots of plants with glabrous leaves growing on medium E, GLE: callus from leaves of plants with glabrous leaves growing on medium E, TRE: callus from roots of plants with tomentose leaves growing on medium E, TLD: callus from leaves of plants with tomentose leaves, growing on medium D, GRC: callus from roots of plants with glabrous leaves growing on medium C, GLD: callus from leaves of plants with glabrous leaves growing on medium D, SS: shoot of a plant grown in soil culture, RS: root of a plant grown in soil culture, SH: shoot of a plant grown in hydroponic culture, RH: root of a plant grown in hydroponic culture, UA—oleanolic acid, OA—ursolic acid, CO: carlina oxide, 3,5QA: 3,5-Di-caffeoylquinic acid, ChA: chlorogenic acid. FC: flavonoid content, DPPH, ABTS and TPC: antioxidant activity evaluated in tests with 2,2-diphenyl-1-picrylhydrazyl, 2-azino-bis (3-ethylbenzthiazoline-6-sulphonic acid and Folin–Ciocalteu reagent, respectively.

**Figure 4 molecules-25-00146-f004:**
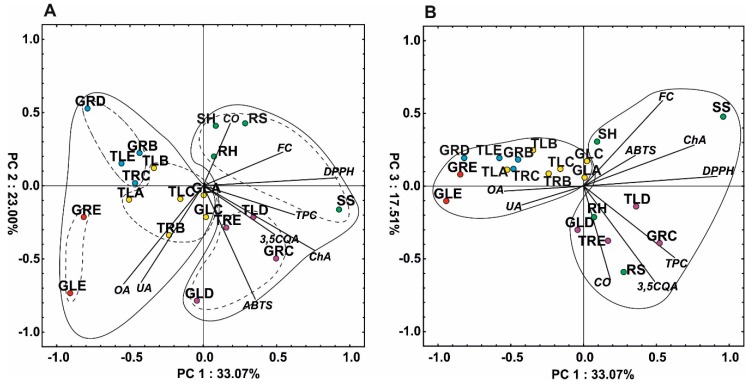
Scatter plot of the PCA of the antioxidant capacity and metabolites determined in the plant material; (**A**) PC1 versus PC2, (**B**) PC1 versus PC3. Different colors describe groups of samples according to the cluster analysis (Figure 3). TRB: callus from roots of plants with tomentose leaves growing on medium B, TLA: callus from leaves of plants with tomentose leaves growing on medium A, TLB: callus from leaves of plants with tomentose leaves growing on medium B, TLC: callus from leaves of plants with tomentose leaves growing on medium C, GLA: callus from leaves of plants with glabrous leaves growing on medium A, GLC: callus from leaves of plants with glabrous leaves growing on medium C, TRC: callus from roots of plants with tomentose leaves growing on medium C, GRB: callus from roots of plants with glabrous leaves growing on medium B, TLE: callus from leaves of plants with tomentose leaves growing on medium E, GRD: callus from roots of plants with glabrous leaves growing on medium D, GRE: callus from roots of plants with glabrous leaves growing on medium E, GLE: callus from leaves of plants with glabrous leaves growing on medium E, TRE: callus from roots of plants with tomentose leaves growing on medium E, TLD: callus from leaves of plants with tomentose leaves, growing on medium D, GRC: callus from roots of plants with glabrous leaves growing on medium C, GLD: callus from leaves of plants with glabrous leaves growing on medium D, SS: shoot of a plant grown in soil culture, RS: root of a plant grown in soil culture, SH: shoot of a plant grown in hydroponic culture, RH: root of a plant grown in hydroponic culture, UA: oleanolic acid, OA: ursolic acid, CO: carlina oxide, 3,5QA: 3,5-Di-caffeoylquinic acid, ChA: chlorogenic acid. FC: flavonoid content, DPPH, ABTS and TPC—antioxidant activity evaluated in tests with 2,2-diphenyl-1-picrylhydrazyl, 2-azino-bis (3-ethylbenzthiazoline-6-sulphonic acid and Folin–Ciocalteu reagent, respectively.

**Table 1 molecules-25-00146-t001:** Effectiveness of callus induction in different variants of Murashige & Skoog Medium.

Plant Morphology	Explant	Abundance of Callus Induced on Medium:				
		A	B	C	D	E
Plants with glabrous leaves	Leaves	++	+++	+++	+++	+
	Roots	++	+	++	++	+
Plants with tomentose leaves	Leaves	++	+++	++	+++	+
	Roots	++	++	++	++	+

Callus: +++ abundant, ++ moderate, + poor; medium: (A) with indole-3-butyric acid (0.5 mg∙L^−1^) and 6-(γ,γ-dimethylallylamino)purine (2 mg∙L^−1^); (B) with indole-3-butyric acid (0.5 mg∙L^−1^) and kinetin (2 mg∙L^−1^); C—with 2,4-dichlorophenoxyacetic acid (1 mg∙L^−1^) and kinetin (1 mg∙L^−1^); D—with 2,4-dichlorophenoxyacetic acid (1 mg∙L^−1^) and 6-benzylaminopurine (1 mg∙L^−1^); E—2,4-dichlorophenoxyacetic acid (2 mg∙L^−1^) and thidiazuron (1 mg∙L^−1^).

**Table 2 molecules-25-00146-t002:** Flavonoid content and antioxidant activity (mg∙g^−1^ dry plant material ± SE) in extracts from the investigated plant materials (*n* = 3).

Medium	Plants with Tomentose Leaves								Plants with Glabrous Leaves								Plants from Soil Cultivation		Plants from Hydroponic Cultivation	
	Calus from the Root			Calus from the Leaf					Calus from the Root				Calus from the Leaf				Roots	Leaves	Roots	Leaves
	B	D	E	A	B	C	D	E	B	C	D	E	A	C	D	E				
**FC**	0.84 ± 0.07 ^d^	0.63 ± 0.09 ^de^	0.85 ± 0.07 ^d^	1.04 ± 0.09 ^cd^	0.42 ± 0.04 ^e^	0.57 ± 0.05 ^e^	0.82 ± 0.07 ^d^	0.99 ± 0.08 ^cd^	0.50 ± 0.05 ^e^	0.70 ± 0.06 ^de^	0.05 ± 0.03 ^f^	2.18 ± 0.04 ^bc^	0.81 ± 0.05 ^d^	0.69 ± 0.04 ^de^	0.77 ± 0.04 ^d^	1.57 ± 0.12 ^c^	2.92 ± 0.37 ^b^	34.67 ± 1.36 ^a^	3.69 ± 0.67 ^b^	38.26 ± 5.50 ^a^
**TPC**	6.46 ± 0.24 ^ef^	4.81 ± 0.13 ^g^	35.89 ± 1.29 ^a^	7.81 ± 0.28 ^de^	1.92 ± 0.09 ^h^	5.90 ± 0.12 ^fg^	19.66 ± 0.97 ^b^	6.74 ± 0.28 ^ef^	2.73 ± 0.05 ^h^	28.05 ± 1.07 ^a^	n. d.	5.31 ± 0.22 ^fg^	13.44 ± 0.34 ^c^	6.46 ± 0.19 ^ef^	10.92 ± 0.26 ^cd^	17.93 ± 0.85 ^b^	18.73 ±1.32 ^b^	19.06 ± 0.30 ^b^	19.90 ±2.54 ^b^	12.05 ± 0.53 ^c^
**DPPH**	6.69 ± 0.12 ^b^	3.87 ± 0.10 ^d^	5.93 ± 0.14 ^bc^	4.18 ± 0.11 ^cd^	4.74 ± 0.13 ^c^	6.37 ± 0.15 ^b^	7.25 ± 0.26 ^b^	2.36 ± 0.05 ^e^	3.57 ± 0.13 ^d^	6.50 ± 0.18 ^b^	0.01 ± 0.01 ^f^	2.18 ± 0.05 ^e^	5.97 ± 0.12 ^bc^	6.46 ± 0.25 ^b^	6.65 ± 0.27 ^b^	2.82 ± 0.11 ^de^	5.24 ±1.26 ^bc^	17.02 ± 2.43 ^a^	4.72 ± 0.62 ^c^	2.71 ± 0.37 ^de^
**ABTS**	13.34 ± 0.50 ^bc^	10.65 ± 0.25 ^de^	13.74 ± 0.59 ^bc^	9.37 ± 0.41 ^ef^	8.56 ± 0.27 ^f^	13.63 ± 0.51 ^bc^	15.20 ± 0.33 ^ab^	10.76 ± 0.27 ^de^	10.37 ± 0.24 ^de^	14.70 ± 0.47 ^ab^	1.12 ± 0.02 ^g^	8.23 ± 0.16 ^f^	10.21 ± 0.22 ^de^	12.75 ± 0.48 ^c^	14.89 ± 0.62 ^ab^	10.51 ± 0.57 ^d^	11.18 ± 0.63 ^cd^	16.99 ± 0.66 ^a^	8.22 ±1.55 ^f^	8.08 ± 0.48 ^f^

FC—flavonoid content, DPPH, ABTS and TPC—antioxidant activity evaluated in tests with 2,2-diphenyl-1-picrylhydrazyl, 2-azino-bis (3-ethylbenzthiazoline-6-sulphonic acid and Folin–Ciocalteu reagent, respectively. Different letters in each line indicate significant differences shown by Tukey’s test at *p* < 0.05.

**Table 3 molecules-25-00146-t003:** Content of determined metabolites (mg∙g^−1^ dry plant material ± SE) in extracts from the investigated plant materials (*n* = 3).

Medium	Plants with Tomentose Leaves								Plants with Glabrous Leaves								Plants from Soil Cultivation		Plants from Hydroponic Cultivation	
	Calus from the Root			Calus from the Leaf					Calus from the Root				Calus from the Leaf				Roots	Leaves	Roots	Leaves
	B	D	E	A	B	C	D	E	B	C	D	E	A	C	D	E				
**ChA**	4.98 ± 0.25 ^d^	1.37 ± 0.19 ^g^	2.26 ± 0.13 ^fg^	2.40 ± 0.12 ^f^	4.46 ± 0.22 ^de^	3.71 ± 0.12 ^e^	6.35 ± 0.36 ^c^	0.13 ± 0.02 ^h^	0.78 ± 0.04 ^g^	8.47 ± 0.41 ^b^	n. d.	0.82 ± 0.05 ^g^	6.42 ± 0.38 ^c^	8.44 ± 0.38 ^b^	6.50 ± 0.22 ^c^	0.78 ± 0.04 ^g^	2.44 ± 0.24 ^f^	10.21 ± 1.69 ^a^	2.16 ± 0.28 ^g^	2.82 ± 0.30 ^ef^
**3,5CQA**	1.31 ± 0.08 ^cd^	0.64 ± 0.05 ^fg^	1.64 ± 0.12 ^bc^	0.52 ± 0.03 ^g^	0.47 ± 0.03 ^g^	1.11 ± 0.07 ^d^	2.02 ± 0.10 ^ab^	0.08 ± 0.03 ^h^	0.50 ± 0.01 ^g^	2.97 ± 0.13 ^a^	n. d.	0.39 ± 0.01 ^g^	1.05 ± 0.08 ^de^	1.15 ± 0.11 ^d^	2.93 ± 0.14 ^a^	0.40 ± 0.01 ^g^	1.57 ± 0.11 ^c^	0.91 ± 0.13 ^ef^	1.78 ± 0.33 ^b^	0.77 ± 0.11 ^f^
**OA**	0.15 ± 0.01 ^de^	0.03 ± 0.04 ^h^	0.05 ± 0.08 ^fg^	0.11 ± 0.01 ^e^	0.03 ± 0.01 ^g^	0.05 ± 0.03 ^g^	n. d.	0.04 ± 0.02 ^g^	n. d.	0.04 ± 0.02 ^g^	n. d.	0.10 ± 0.01 ^ef^	0.04 ± 0.03 ^gh^	0.04 ± 0.02 ^gh^	0.07 ± 0.03 ^f^	0.23 ± 0.01 ^c^	n. d.	2.05 ± 0.08 ^a^	n. d.	1.20 ± 0.53 ^b^
**UA**	n. d.	0.08 ± 0.02 ^d^	0.03 ± 0.01 ^e^	0.05 ± 0.03 ^de^	n. d.	n. d.	n. d.	n. d.	n. d.	0.03 ± 0.01 ^e^	n. d.	0.20 ± 0.02 ^c^	n. d.	0.04 ± 0.02 ^de^	0.28 ± 0.02 ^b^	0.29 ± 0.02 ^b^	n. d.	5.57 ± 0.22 ^a^	n. d.	3.64 ± 0.95 ^a^
**CO**	n. d.	n. d.	n. d.	n. d.	n. d.	n. d.	n. d.	n. d.	n. d.	n. d.	n. d.	n. d.	n. d.	n. d.	n. d.	n. d.	8.73 ± 2.31 ^a^	n. d.	0.14 ± 0.03 ^b^	n. d.

ChA—chlorogenic acid, 3,5-di-caffeoylquinic acid, OA—oleanolic acid, UA—ursolic acid, CO—carlina oxide, n. d.—not detected Different letters in each line indicate significant differences shown by Tukey’s test at *p* < 0.05.

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
