# Peer review of "The Impact of Different Cultivation Systems on the Content of Selected Secondary Metabolites and Antioxidant Activity of Carlina acaulis Plant Material"

_molecules, 2019, doi:10.3390/molecules25010146_

Round 1
Reviewer 1 Report
This paper evaluated the effects of different kind of cultivations on the amounts of some secondary metabolites as well as total phenols and antioxidant activity in C. acaulis plant material. The limited occurrence of this plant for its use in alternative medicine urges the investigation of alternative ways for obtaining this plant material. Thus, in the present paper, in vitro cultures, hydroponic cultures, and field cultivation were performed.
The topic is of interest due to the potential healthy properties of C. acaulis extracts and falls into the scope of the journal.
However, notwithstanding a comparison among the different cultivation systems could be expected, a clear conclusion of the best alternative way to the field cultivation was not drawn.
Results and Discussion as well as conclusion are the sections which mostly suffer of clarity and must be improved. Advantages and disadvantages of each cultivation system should be highlighted, included the possibility to obtain more frequently plant material. The latter should be considered in the discussion of the yield characterizing each cultivation system: soil cultivation allows higher yield but after one–two years of cultivation? Specify.
Discussion is almost descriptive and should be improved considering the relationship between the different environmental conditions of each cultivation system and the possibility to induce secondary metabolites. At this regard some updated literature should be introduced.
Overall the Ms should be improved. In particular:
Lines 31-33: rewrite; Line 80: add references; Line 91: report size of plants cultivated in the field at the end of the first year of growth; Lines 148.151: rephrase. What is the meaning?; Line 158: TPC (antioxidant activity evaluated in tests with Folin–Ciocalteu reagent). Is it a mistake?; Line 169: include literature; Check the use of abbreviations throughout the Ms.: report in full the first time they appear.Author Response
Dear Reviewer 1,
Thank you very much for your valuable comments. The detailed list of responses is given below. The changes made in the text of the revised manuscript are marked in yellow color. We hope that the modifications and explanation will be acceptable for you.
This paper evaluated the effects of different kind of cultivations on the amounts of some secondary metabolites as well as total phenols and antioxidant activity in C. acaulis plant material. The limited occurrence of this plant for its use in alternative medicine urges the investigation of alternative ways for obtaining this plant material. Thus, in the present paper, in vitro cultures, hydroponic cultures, and field cultivation were performed. The topic is of interest due to the potential healthy properties of C. acaulis extracts and falls into the scope of the journal. However, notwithstanding a comparison among the different cultivation systems could be expected, a clear conclusion of the best alternative way to the field cultivation was not drawn.
Reply: In our work we tried to find the alternative for obtaining plant material from natural sites and we tested three different cultivation systems. The field cultivation allowed to obtain the material compared to that from natural sites; however, the other cultivation systems has also some benefits. The conclusion on the advantages of the tested systems has been added.
Results and Discussion as well as conclusion are the sections which mostly suffer of clarity and must be improved. Advantages and disadvantages of each cultivation system should be highlighted, included the possibility to obtain more frequently plant material. The latter should be considered in the discussion of the yield characterizing each cultivation system: soil cultivation allows higher yield but after one–two years of cultivation? Specify.
Reply: The advantages of each tested cultivation system has been described in Introduction and we also added some information in 2 Section. Plants obtained after one year of field cultivation had smaller size than those after 60 days growing in hydroponics. However, after two years the biomass of plant cultivated in soil was significantly higher comparing to 60 days old hydroponic plants – the information explanation has been added to the text.
Discussion is almost descriptive and should be improved considering the relationship between the different environmental conditions of each cultivation system and the possibility to induce secondary metabolites. At this regard some updated literature should be introduced.
Reply: We agree with your suggestion; however, such discussion is difficult because literature on comparison the different cultivation systems in context of plant metabolites is scarce. Moreover, because the time for review is short, we could not to introduce the all suggested improvement. However, we tried to add some essential information and some additional references have been added according to your suggestion.
Overall the Ms should be improved. In particular:
Lines 31-33: rewrite;
Reply: The sentence has been rewritten.
Line 80: add references;
Reply: References have been added.
Line 91: report size of plants cultivated in the field at the end of the first year of growth;
Reply: We were not able to assess the weight of plants after the first year of growth because it would require digging the plants out. In traditional medicine, raw materials obtained in the second year of plant growth are used. To obtain a material corresponding to such a raw material, we were forced to leave the plants in the soil. We could only visually assess the size of the plants in the first year of cultivation. It was comparable to the hydroponic plants we obtained.
Lines 148.151: rephrase. What is the meaning?;
Reply: This has been corrected
The phrase has been reformatted. Line 158: TPC (antioxidant activity evaluated in tests with Folin–Ciocalteu reagent). Is it a mistake?;
Reply: Folin-Ciocalteu reagent has been traditionally used for many years as a measure of total phenolics in natural products, but as the mechanism of detection is based on oxidation/reduction reaction, this method should be considered as a method for assessment of antioxidant capacity (Prior R.L., Wu X., Schaich K. 2005. J. Agric. Food Chem. 53, 4290-4302)
Line 169: include literature;
Reply: The sentence "This finding is in agreement with literature data." has been deleted.
Check the use of abbreviations throughout the Ms.: report in full the first time they appear.
Reply: We have checked the use of abbreviations in the manuscript.
Reviewer 2 Report
The MS deals with the comparison of several cultivation techniques of C. acaulis. Due to the ethnopharmacological importance of the plant and the limitation of its natural presence, this approach could be of great interest. The MS is well written, the experiments well designed. The statistical analysis is robust, even if presentation is not easy for non experts in the field.
In my opinion the major lack of the MS is the absence of any biological test (antimicrobial for instance), that would have significantly improved the comparison and thus the MS. However, the data reported highlight the substantial similarity between hydroponic and field cultivation. This evidence alone is an important achievement that could help to increase the utilization of this plant.
So I suggest to accept the MS after some minor modification.
MAJOR POINTS: Why did the Authors choose methanol as the only extraction solvent? Maybe this choice should be justified in the text. In any case, all the data data reported can be also be extended on the methanol-extractable portion of the plant. This should be highlighted and stressed all along the MS. Unless the Authors are able to provide data about the fact the ChA, 3,5CQA, OA and UA can be totally extracted by methanol and not by other solvents.
In the HPLC chromatograms, did the Authors only find these metabolites of there were other (not quantified) chemicals present in smaller amounts? In case, did the Authors show any difference among the samples?
MINOR POINTS:
Line 78: C. acaulis in italics.
Line 80: idem.
Line 82: “”In this study, we have attempted to obtain the plant material” this sentence seems to be either incomplete or out of context.
Lines 148-151: is this an abbreviations list? Maybe it should not be place here.
Author Response
Dear Reviewer 2,
Thank you very much for your valuable comments. The detailed list of responses is given below. The changes made in the text of the revised manuscript are marked in yellow color. We hope that the modifications and explanation will be acceptable for you.
The MS deals with the comparison of several cultivation techniques of C. acaulis. Due to the ethnopharmacological importance of the plant and the limitation of its natural presence, this approach could be of great interest. The MS is well written, the experiments well designed. The statistical analysis is robust, even if presentation is not easy for non experts in the field.
In my opinion the major lack of the MS is the absence of any biological test (antimicrobial for instance), that would have significantly improved the comparison and thus the MS. However, the data reported highlight the substantial similarity between hydroponic and field cultivation. This evidence alone is an important achievement that could help to increase the utilization of this plant.
Reply: Thank you very much for recognition and kind words about our research. We are currently conducting pharmacological studies of pure compounds isolated from Carlina plants, but they are part of separate projects.
So I suggest to accept the MS after some minor modification.
MAJOR POINTS: Why did the Authors choose methanol as the only extraction solvent? Maybe this choice should be justified in the text. In any case, all the data data reported can be also be extended on the methanol-extractable portion of the plant. This should be highlighted and stressed all along the MS. Unless the Authors are able to provide data about the fact the ChA, 3,5CQA, OA and UA can be totally extracted by methanol and not by other solvents. In the HPLC chromatograms, did the Authors only find these metabolites of there were other (not quantified) chemicals present in smaller amounts? In case, did the Authors show any difference among the samples?
Reply: We chose methanol as a relatively universal extractant. Our previous unpublished research shows that three times ultrasound-assisted extraction, while maintaining the ratio of the raw material weight to the methanol volume specified in the methodology, is exhaustive. Methanol is an extractant in which both chlorogenic acids and more non-polar triterpenes and carlin oxide are dissolved. We agree with the reviewer that other better solvents for the extraction of single compounds can be selected; however, we have found that methanol is a good extractant for comparative assessment of plant material.
MINOR POINTS:
Line 78: C. acaulis in italics.
Reply: This has been corrected
Line 80: idem.
Reply: Thank you for your comment. The text has been corrected in accordance with the Reviewer's recommendations.
Line 82: “”In this study, we have attempted to obtain the plant material” this sentence seems to be either incomplete or out of context.
Reply: The sentence has been deleted.
Lines 148-151: is this an abbreviations list? Maybe it should not be place here.
Reply: This fragment of the text has been included in the description of the figure.
Reviewer 3 Report
The paper describes the effects of different cultivation systems on the
amount of some secondary metabolites and antioxidant activity of Carlina acaulis (Asteraceae). Some improvements should be done in the paper such as:
Lines 78 and 80: C. acaulis should be written in italics as well as the plant names in references.
Line 79: Authors affirm that in a previous investigation higher amounts of four components were found in the plant. However, the respective reference was not mentioned.
Lines 148-151: this text fragment should be incorporated in the figure caption.
Line 159: statistically
In the 5th paragraph of page 8 as well as in the 3.6. item, the reference (Strzemski et al. 2016) should be corrected.
Figure 3 and the respective caption must be presented in the same page.
What is the reason for no production of polyacetylene in calluses? Additionally, why the amount of polyacetylene in plants from soil was higher? Both questions should be answered in the text.
PCA shown in Figure 4 is difficult to understand. I suggest to authors use colors to differentiate groups.
Figure S1. Although obvious, the caption must indicate the plant that grew in soil and water.
Author Response
Dear Reviewer 3,
Thank you very much for your valuable comments. The detailed list of responses is given below. The changes made in the text of the revised manuscript are marked in yellow color. We hope that the modifications and explanations will be acceptable for you.
The paper describes the effects of different cultivation systems on the amount of some secondary metabolites and antioxidant activity of Carlina acaulis (Asteraceae). Some improvements should be done in the paper such as:
Lines 78 and 80: C. acaulis should be written in italics as well as the plant names in references.
Reply: Thank you for your comment. The text has been corrected in accordance with the Reviewer's recommendations
Line 79: Authors affirm that in a previous investigation higher amounts of four components were found in the plant. However, the respective reference was not mentioned.
Reply: The references have been added.
Lines 148-151: this text fragment should be incorporated in the figure caption.
Reply: This fragment of the text has been included in the description of the figure.
Line 159: statistically
Reply: This has been corrected.
In the 5th paragraph of page 8 as well as in the 3.6. item, the reference (Strzemski et al. 2016) should be corrected.
Reply: This has been corrected.
Figure 3 and the respective caption must be presented in the same page.
Reply: This has been corrected.
What is the reason for no production of polyacetylene in calluses? Additionally, why the amount of polyacetylene in plants from soil was higher? Both questions should be answered in the text.
Reply: These issues are discussed on page 8 as recommended by the reviewer.
PCA shown in Figure 4 is difficult to understand. I suggest to authors use colors to differentiate groups.
Reply: This has been corrected.
Figure S1. Although obvious, the caption must indicate the plant that grew in soil and water.
Reply: This has been corrected.
Round 2
Reviewer 1 Report
Notwithstanding some modifications to the Ms have been performed, differences among the different kind of cultivations have not been sufficiently highlighted to reach a clear conclusion of this study.
Author Response
Dear Reviewer,
Thank you very much for your valuable comments. According to your suggestions, some information has been added to the introduction and discussion. The changes made in the text of the revised manuscript are marked in yellow color. We hope that the modifications and explanation will be acceptable for you.